# Palliative care needs and preferences of female patients and their caregivers in Ethiopia: A rapid program evaluation in Addis Ababa and Sidama zone

**Mirgissa Kaba**[1]*, **Marlieke de Fouw**[2], **Kalkidan Solomon Deribe**[1]*, **Ephrem Abathun**[3], **Alexander Arnold Willem Peters**[2], **Jogchum Jan Beltman**[2]

1 Department of Preventive Medicine, School of Public Health, College of Health Sciences, Addis Ababa University, Addis Ababa, Ethiopia, 2 Department of Gynecology, Leiden University Medical Center, Leiden, The Netherlands, 3 Hospice Ethiopia, Addis Ababa, Ethiopia

* mirgissk@yahoo.com (MK); kallkidansolomon@gmail.com (KSD)

**Data Availability Statement:** All relevant data are within the manuscript and its supporting information files.

## Abstract

### Introduction

In Ethiopia there is an extensive unmet need for palliative care, while the burden of non-communicable diseases and cancer is increasing. This study aimed to explore palliative care needs and preferences of patients, their caregivers, and the perspective of stakeholders on service provision in palliative programs for women, mostly affected by cervical cancer and breast cancer.

### Methods

A rapid program evaluation using a qualitative study approach was conducted in three home-based palliative care programs in Addis Ababa and Yirgalem town, Ethiopia. Female patients enrolled in the programs, and their primary caregivers were interviewed on palliative care needs, preferences and service provision. We explored the views of purposely selected stakeholders on the organization of palliative care and its challenges. Audio-taped data was transcribed verbatim and translated into English and an inductive thematic analysis was applied. Descriptive analyses were used to label physical signs and symptoms using palliative outcome scale score.

### Results

A total of 77 interviews (34 patients, 12 primary caregivers, 15 voluntary caregivers, 16 stakeholders) were conducted. The main physical complaints were moderate to severe pain (70.6%), followed by anorexia (50.0%), insomnia, nausea and vomiting (41.2%). Social interaction and daily activities were hampered by the patients' condition. Both patients and caregivers reported that programs focus most on treatment of symptoms, with limited psychosocial, emotional, spiritual and economic support. Lack of organizational structures and policy directions limit the collaboration between stakeholders and the availability of holistic home-based palliative care services.

**Funding:** The study was supported by the Ethiopia Female Cancer Initiative (EFCI) project of Cordaid Ethiopia, and Treub Foundation from The Netherlands.

**Competing interests:** EA is executive director of Hospice Ethiopia. MF and JJ provided training and supervision of cervical cancer screening activities via the Female Cancer Foundation for the EFCI program. The authors declare they have no competing interests. This does not alter our adherence to PLOS ONE policies on sharing data and materials.

**Abbreviations:** ART, Anti-Retroviral Treatment; FC, Family caregiver; FMOH, Federal Ministry of Health; HIV/AIDS, Human Immune Deficiency Virus/ Acquired Immune Deficiency Syndrome; IQR, Interquartile range; NGO, Non-governmental organization; POS, Palliative outcome scale; SPSS, Statistical packages for social science; WHO, World health organization.

## Conclusions

Although female patients and caregivers appreciated the palliative care and support provided, the existing services did not cover all needs. Pain management and all other needed supports were lacking. Multi-sectorial collaboration with active involvement of community-based structures is needed to improve quality of care and access to holistic palliative care services.

## Introduction

In Ethiopia there is an extensive unmet need for palliative care, while the burden of non-communicable diseases and cancer is increasing [1]. Access to palliative care is a human right although there are evident disparities in its provision [2, 3]. In the past decade, very few of the people in need of palliative care across the globe receive it, and referral to palliative care teams for most patients occurs in the last 2 to 6 months of life or not at all [4, 5]. In Sub-Saharan Africa where life expectancy is short, supportive and palliative care for severely ill patients are hardly available. Referral to palliative care early in the course of illness is important for optimal quality of life, and at the same time reduces unnecessary hospitalizations and crowding of health-care services [6–8].

Ethiopia is one of the Sub-Saharan countries where high burden of suffering and lack of access to pain relief and palliative care are apparent. In this paper we assess palliative care and support programs for women, mostly affected by cervical cancer and breast cancer, as integrated part of screening activities. Breast and cervical cancer are the leading cancers among women in Ethiopia, with an annual crude incidence rate of 29.8 and 16.3 per 100.000 respectively [7]. The coverage of prevention programs in Ethiopia is increasing but still limited in terms of components of the program and accessibility for all women at risk. When presented in early stage cervical and breast cancer can be treated with surgery, radiotherapy or chemotherapy. However, most women identified with cervical and breast cancer present in advanced stage when curative treatment is no longer an option [9]. These women and their caregivers should receive support and appropriate care to address their needs, but comprehensive palliative care services are barely accessible.

Despite investments and programs of the Federal Ministry of Health in Ethiopia, the World Health Organization (WHO) and several non-governmental organizations (NGOs), palliative care services and data on palliative care needs in Ethiopia are still lacking or [10, 11]. A study conducted in Addis Ababa showed that 65% of cancer patients admitted in a tertiary referral hospital, the majority with advanced stage disease, did not receive adequate pain management [12]. Untreated pain and high costs associated with life-limiting illness are reported to be main factors leading to psychosocial distress and financial crisis [13]. The Ethiopian Ministry of Health has developed a national guideline on palliative care, but the gap with current service provision is evident [14].

Our study aimed to explore palliative care needs and preferences of female patients with breast and cervical cancer or other life-threatening chronic illnesses, and their primary and voluntary caregivers in three home-based palliative care programs in Ethiopia. Furthermore, we intended to assess the perspectives of stakeholders on the existing service provision and its challenges. Based on our findings we present recommendations to improve and prioritize palliative care provision.

## Methods

### Study design

A rapid evaluation methodology (REM) using a qualitative study approach was conducted in three home-based palliative care and support programs. The rapid evaluation methodology (REM) was developed by WHO to evaluate the performance of health care programs and identify problems in order to develop recommendations for future programming [13]. This methodology was tested in several low- and middle-income countries and was used for evaluation of a palliative care program in Malawi [14].

### Study area and period

The study was conducted in May and June 2018 in Addis Ababa, capital city of Ethiopia, and Yirgalem town, in Sidama region, 320 km south of Addis Ababa. In Addis Ababa the palliative care programs of Hospice Ethiopia and Mary Joy Development Association (MJDA) were assessed, in Yirgalem town the palliative care program of Beza for Generation (B4G). At the time of conducting this study, these programs were to the best of our knowledge the only home-based palliative care and support programs in cancer care in Ethiopia.

Hospice Ethiopia is a NGO with both facility-based and home-based palliative care provision, trained by palliative care specialists from Uganda and Kenya. Mary Joy Development Association and Beza for Generation are local community-based NGOs providing support to palliative patients as part of the Ethiopian Female Cancer Initiative (EFCI), a cervical cancer and breast cancer prevention program, managed by Cordaid Ethiopia and supported by the Female Cancer Foundation from the Netherlands.

### Study population

Looking into the total case population of each program (approximately 80–120 enrolled patients per program) of which the majority fulfilled the inclusion criteria, combined with reaching illustrative sample of patients and caregivers for this rapid evaluation methodology 10–15% of the total case load of each program and a matching number of caregivers (8–12 patients per setting) were interviewed. However, with 8–12 participants from the respective sites, saturation was achieved after the fifth participant was interviewed. Additional participants were interviewed after presumed saturation was achieved to ascertain repetition of evidence.

The primary caregiver of the interviewed patients who were, either relatives to the patient or volunteer were, included in the study. The number of caregivers was lower than for patients, because not all caregivers were present at the time of the scheduled interview or did not provide informed consent. All interviews were conducted individually, with the patient or caregiver, and the researcher. In Yirgalem a translator was present as a third person when necessary.

In addition to patients and their care givers, staff members responsible for palliative care at the Ministry of Health, palliative care providers at facility and community levels program managers of the NGOs involved in palliative care services, community and religious leaders were participated in the study.

### Data collection tool and process

We employed the rapid program evaluation methodology different data sources: 1. Patient files and project reports; 2. Patients and their primary or voluntary caregivers; 3. Key stakeholders from both government, NGOs and the local communities.

Patient files and project reports were used to extract socio-demographic characteristics (age, marital status, educational level), clinical characteristics (diagnosis, HIV status, co-morbidities, medication used) and time of involvement in the program using a standardized form, see S1 Appendix.

For interviews with patients, caregivers and stakeholders open-ended interview guides were developed, based on a study by Herce et al [14] on palliative care in Malawi, see S2-S4 Appendices. The interview guide for patients collected information on socio-demographic characteristics, physical signs and symptoms using the adjusted Palliative Outcome Scale (POS) validated for the African setting [15], and perceived challenges and preferences for palliative care and end-of-life planning.

The interview guide for caregivers collected information on socio-demographic characteristics, perceived support, challenges and preferences for provision of palliative care (S3 Appendix). The interview guide for stakeholders included questions on existing health service activities, and gaps and challenges in palliative care programs (S4 Appendix). All interview guides included questions about the definition and perceived components of palliative care.

The interview guides were prepared in English and translated to Amharic language and Sidama language by an official translator. The translation was cross-checked by two health care professionals and a person without medical background.

The interviews were conducted by six data collectors. In Addis Ababa the team consisted of three Amharic speaking students (two females, one male) of the Master program of Public Health of Addis Ababa University. In Sidama zone the interviews were conducted by three data collectors (two females, one male) fluent in both Amharic and Sidama language, with at least a first degree in a health-related field. All data collectors were recruited based on their experience in social and medical research. All data collectors were trained on the objective of the study, the research protocol including the tools and procedures of data collection for two days by the local principal investigator [MK]. The interviews took place either in the participants' home or at the Hospice Ethiopia health center, depending on their preference, with only the participant, the interviewer and if needed the translator present. Interviews with key stakeholders took place in their workplace or at home, depending on their preference. The interviews were tape-recorded and field notes were taken by the interviewer. Each interview was planned for approximately 40 minutes.

## Data management and analysis

Descriptive analyses were used to label physical signs and symptoms using POS score. We used two categories to express the severity of symptoms; POS score 0 to 2 indicating 'none to mild' complaints, 'POS score 3 to 5 indicating 'moderate to severe' complaints [16] [MK, MF, KS].

The audio-recorded interviews were transcribed in Amharic, translated into English, and aligned with field notes of the interviewer. Inductive approach was used and thematic analysis was applied to the transcribed interviews. Data were coded by two independent researchers [MK, KS] and in case of discrepancies between the two by another researcher [MF] to verify and reach decision. Before the analysis, consensus was reached among the researchers [MK, KS, MF] on the coded themes and subthemes. Data analysis was facilitated by Open-code version 4.02.

## Data quality assurance

A standardized data collection form was developed to extract data from patient files and project reports. In addition to the initial training, data collectors were closely supervised by the

local principal investigator. Interviews were conducted in settings preferred by the patient, caregiver or stakeholder, to ensure a comfortable environment for discussion. The interviewers did not have a relationship with the participants, nor were involved in the palliative care programs. The interviewer validated the obtained information with the participant after the interview, to ensure that the answers were rightly captured.

### Ethics statement

The study was reviewed and approved by the Research and Ethics committee of the School of Public Health, Addis Ababa University, and registered with number prv/154/10. The health authorities of the research sites provided permission for the study. An official letter of permission was provided to the administrative office of each of the selected palliative care centers. Before data collection, the study participants were informed about the purpose of the study and that their decision about participation would not influence the care they received.

For both patients and their caregivers written or oral informed consent was obtained to carry out the interview. Those who gave oral consent did so in presence of a witness in their own language (Amharic in Addis Ababa, Amharic or Sidama in Yirgalem). An appointment was scheduled for an interview once consent was obtained.

During the interview voice recording was made after securing permission. Information obtained was kept confidential, anonymous and used only for this research purpose. After transcribing, the audio-tapes were deleted and hardcopies of the interview were stored at a secured place at Addis Ababa University accessible to the principal investigator.

## Results

A total of 77 in-depth interviews were conducted; 34 interviews with patients, 12 with primary caregivers, 15 with voluntary caregivers and 16 with stakeholders.

### Socio-demographic and disease related characteristics

Table 1 demonstrates the socio-demographic and disease-related characteristics of the patients. The age of the patients ranged from 23 to 80 years with a median age of 47 years. The educational level was low, half of the patients were illiterate (n = 18, 53%) and one third completed primary school (n = 12, 35%). The majority of patients (n = 25, 73%) was unemployed or unable to work.

All patients were tested for HIV/AIDS, 13 (38%) were HIV-positive and therefore enrolled in the palliative care program. Two-third of HIV-positive patients (n = 9, 69%) used antiretroviral treatment [17].

Table 2 illustrates that almost all caregivers (n = 26, 96%) were female and literate, 19 (70%) were employed in governmental and non-governmental organizations. Out of 12 caregivers that were not volunteering with NGOs, 9 (75%) were close relatives to the patients, 3 (25%) were neighbors.

### Palliative Outcome Scale score

Table 3 illustrates POS scores for patients. The complaints that were experienced as moderate to severe by patients were pain (n = 24, 71%), anorexia (n = 17, 50%), insomnia (n = 15, 41%) and nausea and vomiting (n = 14, 41%).

The majority of patients (n = 29, 85%) did not talk with relatives about their condition, were very worried about their condition (n = 22, 65%) and their condition strongly affected activities of daily life (n = 20, 59%) and their social interaction (n = 20, 59%).

**Table 1. Socio-demographic characteristics of patients (n = 34).**

| Socio-demographic characteristics | Frequency | Percentage (%) |
|---|---|---|
| *Marital status* | | |
| Single | 2 | 6 |
| Married | 12 | 35 |
| Separated/divorced | 9 | 27 |
| Widowed | 11 | 32 |
| *Religion* | | |
| Orthodox | 28 | 82 |
| Christian other than orthodox | 5 | 15 |
| Muslim | 1 | 3 |
| *Level of education* | | |
| Illiterate | 18 | 53 |
| Literate | 16 | 47 |
| *Employment status* | | |
| Unemployed | 13 | 38 |
| Work at own home or farmland | 2 | 6 |
| Daily labourer, unskilled/skilled | 7 | 21 |
| Unable to work due to illness | 12 | 35 |
| *Livelihood supports daily expenses (n = 9)* | | |
| Somewhat but other source of income needed | 1 | 11 |
| No | 8 | 89 |
| *Type of diagnosis* | | |
| HIV/AIDS | 13 | 38 |
| Cervical cancer | 10 | 29 |
| HTN & DM | 7 | 21 |
| Breast cancer | 4 | 12 |
| *Tested for HIV* | | |
| Yes | 34 | 100 |
| No | 0 | 0 |
| *HIV-positive patients using ARVs (n = 13)* | | |
| Yes | 9 | 69 |
| No | 4 | 31 |
| Years since diagnosis of HIV/AIDS: median | 6 | |

SD: standard deviation, HIV: Human Immunodeficiency Virus, AIDS: acquired Immunodeficiency Syndrome, HTN: hypertension, DM: diabetes mellitus, ARV: anti-retroviral treatment.

## Thematic analysis

We identified the following themes in the semi-structured interview transcripts: Awareness of palliative care, Organization of palliative care and referral pathways, Current palliative care activities, Physical and psychological impact, End-of-life planning, Preferences for home-based or institutional care and Challenges in palliative care provision.

**Awareness of palliative care.** The meaning of 'palliative care' was perceived differently among the study participants, as illustrated in Box 1. Almost all health professionals described palliative care as care given for terminally ill patients to alleviate their pain and improve the patient's quality of life. Yet, these professionals were not aware of the national guideline on palliative care services in Ethiopia.

**Table 2. Socio-demographic characteristics of caregivers (n = 27).**

| Socio-demographic characteristics | Frequency | Percentage (%) |
|---|---|---|
| *Level of education of the caregiver* | | |
| Illiterate | 1 | 4 |
| Literate | 26 | 96 |
| *Employment status of the caregiver* | | |
| Unemployed | 8 | 30 |
| Employed | 19 | 70 |
| *Caregiver—patient relationship* | | |
| Close relatives | 9 | 33 |
| Neighbors | 3 | 11 |
| Volunteers | 15 | 56 |

SD: standard deviation.

Most patients and caregivers reported that they "never heard" of palliative care and "don't understand" what palliative care is. Few patients and stakeholders other than health professionals, described palliative care as helping patients, elders and orphans who are unable to care for themselves either due to illness or other reasons. Most stakeholders mentioned health care at facilities, home to home visits and advice and provision of economic support as elements of palliative care.

**Organization of palliative care and referral pathways.** The elements of palliative care provided by the organizations involved in this study were not uniform. Hospice Ethiopia provided both home-based support comprising of financial support and provision of analgesics, and outpatient care in the Hospice center which included medical, psychosocial, and financial support and daycare activities. Trained nurses and community volunteers provided palliative care services, while actively involving family members of the patient in care provision. Hospice Ethiopia referred its clients for advanced medical care to Black Lion Hospital. Black Lion Hospital, St Paul's hospital and Yekatit Hospital in Addis Ababa provided palliative care services, but none of the hospitals had inpatient hospice care or home-based palliative care programs.

Unlike Hospice Ethiopia, MJDA and B4G did not have formal referral linkages with health facilities and focused more on home-based supportive care and less on medical care provision. The supportive care program consisted of periodical provision of materials like cloth and food items, financial support aimed at supporting patients for their medical expenses, and drug provision to alleviate pain by trained nurses, both at the organizations' center and at the patients' home. While there were no officially trained palliative care providers within MJDA and B4G, both organizations built on their experience from home-based care to people living with HIV/AIDS and caregivers received basic training in palliative care and support.

Patients were referred to the palliative care programs via health institutions and community volunteers, or patients themselves visited the program centers to apply for the services. Patients needed a referral letter from a health institution stating their diagnosis, before enrollment in the program. Patients, and to some extent families of these patients, orphans and elders, were supported in the programs.

Patients, caregivers and stakeholders stated that health professionals are responsible to screen the patient, treat and refer to other health facilities if indicated. Close relatives and community volunteers were identified as primary care providers. Religious local organizations called 'Idirs' were frequently mentioned to provide financial support to bedridden women, religious leaders provided psychosocial and spiritual support.

**Table 3. Ratings of patients' signs and symptoms using the adjusted African Palliative Outcome Scale (POS).**

| Signs and symptoms | Frequency | Percentage |
|---|---|---|
| *Pain* | | |
| None-mild | 10 | 29.4 |
| Moderate-severe | 24 | 70.6 |
| *Nausea and Vomiting* | | |
| None-mild | 20 | 58.8 |
| Moderate-severe | 14 | 41.2 |
| *Constipation* | | |
| None-mild | 23 | 67.6 |
| Moderate-severe | 11 | 32.4 |
| *Diarrhea* | | |
| None-mild | 30 | 88.2 |
| Moderate-severe | 4 | 11.8 |
| *Anorexia (trouble in eating)* | | |
| None-mild | 17 | 50.0 |
| Moderate-severe | 17 | 50.0 |
| *Coughing* | | |
| None-mild | 25 | 73.5 |
| Moderate-severe | 9 | 26.5 |
| *Trouble in breathing* | | |
| None-mild | 28 | 82.4 |
| Moderate-severe | 6 | 17.6 |
| *Insomnia (trouble in sleeping)* | | |
| None-mild | 19 | 58.8 |
| Moderate-severe | 15 | 41.2 |
| *Worried about their health* | | |
| None-mild | 12 | 35.3 |
| Moderate-severe | 22 | 64.7 |
| *Sharing with family or friends about their health* | | |
| None-mild | 29 | 85.3 |
| Moderate-severe | 5 | 14.4 |
| *Daily activities affected* | | |
| None-mild | 14 | 41.2 |
| Moderate-severe | 20 | 58.8 |
| *Social interaction affected* | | |
| None-mild | 14 | 41.2 |
| Moderate-severe | 20 | 58.8 |

None-mild = POS score 0, 1 or 2, and Moderate-severe = POS score 3, 4 or 5.

**Current palliative care activities.** More than two third (n = 27) of the patients claimed to have received medical, psychosocial and financial support from palliative care providers. Yet, all patients complained that the support they received was not sufficient. It was hard to specify this need in detail, although repeated reference was made to persisting pain and failure to make their living (Box 2).

The problem of insufficient palliative care provision was well-recognized at Ministry of Health level, related to insufficient budget, human resources and attention for palliative care services.

Box 1. Awareness and organization of palliative care

*Awareness about palliative care*

*"I never heard about palliative care. I usually see people coming to the Hospice center and thought these are people who do not have support and come to the center to seek support."* (35 year old, female, HIV-positive patient)

*"We know death is an inevitable event but when we try to advice patients not to lose hope and be ready for that, I think this is useful for the patient. That is probably an important care although this is not widely known and available to all patients."*

(63 year old, male, religious leader)

*"I think it is giving a home to home care for patients with severe sickness which is non curable. Through this service their pain could be relieved to some extent so that they may pass without pain."* (26 years old, female, volunteer)

*Organization of palliative care*

*"Support to a patient who is seriously sick is often a family and community affair. At household level family members support on a daily basis to meet the demand of the patient. Community members also visit and offer advice and encouragement. Health facilities, in my view, do not help with social, spiritual and economic demands of the patient. They are responsible only for routine health service provision."* (30 year old, female, volunteer)

*"As a religious person we provide spiritual support to sick people, and health facilities provide health care. Both services are meant to improve the quality of life of the patient."*

(63 year old, male, religious leader)

Box 2. Current palliative care activities

*"I received drugs and some money from Hospice. However, I want to recover fully from my illness and get back to my routines. I need more support that may help me full recovery and support to my children."* (70 years old, female, congested heart failure patient)

*"She (the volunteer provider) has been caring. However, I am not happy and lose hope when my pain comes back. I then feel uncertain about my life. I feel like am dying. It is bad to live under uncertainty, losing your ability to make decision about myself. The volunteer at times fails to help under such circumstances."*

(42 years old, female, chronic kidney disease patient).

*"There is a long way to go to improve palliative care service in Ethiopia. There is no independent case team responsible to coordinate palliative care within the Ministry of Health or in regional health offices. As a result, there is no budget allocated for the program, no formally trained human resources and most importantly this is not given as much attention as other programs. In general, for me it is very difficult to say that there is palliative care as holistic as it should be."*

(45 year old, female, palliative care focal person) MOH)

Box 3. Physical and psychological impact

*"I was very sick and couldn't do my routine activities. So, I quitted my job because of the illness plus I have to now become dependent on someone else. . . I am hopeless and sad. I am however getting support from Hospice Ethiopia which lessens the tension I am in."*

(58 year old, female, cervical cancer patient)

*"I bleed every time. It clots and clots and brought offensive smell since I do not have support to clean it and of course no one comes closer. I got weaker and weaker. Only recently volunteers from Beza came to help me–thank God."*

(38 year old, female, HIV-positive patient)

*"I enclose myself in the house because people tell me quite indirectly that I stink. Because of this, I always cry and wish I could kill myself."*

(52 year old, female, hypertensive and diabetic patient)

*"When you are largely dependent on others, you feel to be valueless. That compels you at times to wish death the soonest. What should I do? You know what; I would love to die to get away from this suffering."* (46 year old, female, cervical cancer patient)

*"I suffered from the disease that restricted my movement. I have severe cough and accompanying pain of my abdomen and nausea. Although I was told there is no treatment, I can't pay for better medical service. This makes me sad and feel worthless. At times I ask myself what mistake did I commit for I feel this is nothing but punishment."*

(77 years old, female, breast cancer patient)

*"Following regular visits by the priest, I do regular prayer and got much stronger inside. I am also using holy water at home now nearly for a year. I am feeling much better with my health."* (30 year, female, HIV-positive and skin cancer patient)

**Physical and psychological impact.**   All patients reported panic, anxiety and sadness when they heard their diagnosis. Most patients recalled they refrained from their daily activities and social interactions.

Patients expressed their perception of their disease with words like "painful", "bad disease" or "disgusting". Women who were bedridden due to advanced cervical cancer reported to have experienced continuous, aching, foul smelling vaginal bleeding, and several women with different diagnoses reported severe pain to the extent of difficulty in breathing.

All patients experienced physical pain, psychosocial or emotional grief and spiritual neglect.

Patients agreed that spiritual support from religious leaders gave hope in their situation, although the religious leaders did not receive adequate training within the palliative care programs.

**End-of-life planning.**   The majority of participants were not aware what their disease meant in terms of survival, and most patients (n = 25) did not have an end-of-life plan. The volunteer providers at community level also did not recognize these implications (Box 4).

**Preferences for institutional or home-based care.**   Half of the patients (n = 18) preferred to stay at home and be cared for at home. Their preference was based on having company from family members, unlike in health facilities where patients would be lonelier. Some

---

> ## Box 4. End-of-life planning
>
> *"No we do not have an end-of-life plan. . .Because we thought that planning about end-of-life is interfering with the work of God."* (40-year-old, female, volunteer)
>
> *"I pay for Idir and church because this is what everyone does and it is meant to ensure easy burial. I don't plan for end-of-life because I wish to live a healthy life."*
>
> (65 year old, female, breast cancer patient)
>
> *"Since I believe that God helps me; I don't die and get separated from people I love and live together. I keep praying believing the Almighty will save me. So, I fight my disease and want survive longer."*(58 year old, female, cervical cancer patient)

patients mentioned that health care staff was not respectful to patients and would not care adequately for their symptoms like pain.

Other patients preferred provision of palliative care in health facilities in order to be in close proximity to medical care when needed and to have more privacy than at home (Box 5).

**Challenges in palliative care provision.**   The collaboration between potential stakeholders and their respective roles and responsibilities were not well defined. Providers felt that there is no clear guideline on palliative care provision, while patients and caregivers were looking for more support.

Providers at different levels reported challenges in palliative care provision. The most common challenges included lack of awareness among community and facility level providers, lack of guidance for care providers, lack of a structure that clarifies roles and expectations at different levels, lack of accountability and poor commitment of health care staff to palliative care programs.

At community level, volunteers reported to have limited information about the service. The shortage of volunteers and severity of the disease distressed the volunteers who are willing to care more for their patient. Patients who are bedridden revealed to suffer from dwindling livelihood, lack of appropriate information about their status and lack of support in relieving pain when needed.

---

> ## Box 5. Preference for institutional or home-based care
>
> *"I prefer to stay at home. I prefer to be with my families. There are organizations and health centers which asked to take and care of me in their institution but I refused them. I fear to be alone there in the hospital."*(47 years old, female, HIV-positive patient)
>
> *"Care at home is much better, for family members sympathize and give me much care. In the health facilities, professionals are not respectful and do not show any sympathy and do not care much for the pain I suffer from"* (80 years old, female, arthritis patient)
>
> *"I prefer to get care at health facilities. This will minimize the number of people that visit me at home and give me difficult time to answer different questions that at times are annoying"* (42 years old female, HIV-positive patient)

---

Box 6. Challenges in palliative care

*"In as much as palliative care improves the quality of life of a terminally sick person, the service is not as holistic as it should be and there is no line of accountability at different levels. For me mere focus on pain management and provision of financial support which is not sustainable, is not wise."* (38 year old, male, palliative care focal person)

*". . .When we compare administrative support even the Ministry of Health didn't give much focus to palliative care. This can be explained in terms of lack of budget and necessary training or man power. As I told you we are in establishing the palliative care unit but we face a lot of challenges since the administrators of the hospital are not that much dedicated to this service."* (42 year old, male, palliative care focal person)

*"To me the major challenge is the non-supportive attitude and poor commitment of providers especially at facility level. I witnessed that professionals at facility level are not well prepared to help patients with non-curable diseases, including how they break the bad news is unprofessional. They don't care much about the terminally ill patients. They do not know grief counseling and how to support patients."*

(38 year old, male, hospital palliative care focal person)

*"As volunteers, myself and my friends involve in this are very happy to care for bedridden patients. However, we do not have relevant information on what we should do and should not do. Often the suffering of the patient is so consuming that some of us get even sick. Besides, we do not have protective supplies such as gloves, so that we get worried."*

(42 years old, female, volunteer)

## Discussion

Our study focused on the perceived needs and preferences of both patients, caregivers and stakeholders in home-based palliative care programs, and is one of the first studies to be conducted about this topic in Ethiopia. We found that awareness of palliative care was limited and several challenges need to be addressed; insufficient medical and psychosocial support to address patients' complaints like severe pain, anorexia and anxiety in a holistic approach, lack of support for palliative care providers and caregivers to cope with their emotionally challenging task, lack of collaboration between stakeholders with a need to define roles and responsibilities, and create awareness and ownership at both community, healthcare and policy level to make palliative care a priority.

### Awareness of palliative care

Our study showed poor awareness of palliative care among all participants at both policy level, health facility level and community level. Health care professionals were not familiar with the Ethiopian national guideline, patients and caregivers were not aware of the implications and existing structures of palliative care. Hence it is difficult for patients and their caregivers to use a service of which they are not well aware, and for care providers to provide an adequate level of palliative care. This situation is not limited to Ethiopia, but was found in women living in other parts of Africa, Asia, Eastern-Europe and the United States as well [5, 18–21]. Limited awareness highlights the need for improved understanding of what palliative care means and whom it can benefit.

The lack of awareness of palliative care draws a parallel to the care for HIV/AIDS patients during the early years of the epidemic. It has taken many years of ongoing investment at all levels, including strong awareness programs at community level, to integrate HIV/AIDS care in routine health care activities and to make it accessible at health center level. Experiences from the time when HIV incidence was increasing plays an important role in rolling out palliative care services. For the two community-based organizations involved in our study, MJDA and B4G, the home-based care service they provided for people living with HIV/AIDS provided a useful foundation for the palliative support program. Still, the level of care currently provided proved insufficient to meet patients' needs. and an organized effort at national and regional level is needed to develop norms and guidance for palliative care provision. To improve awareness at community level, coffee ceremonies have proven to be a context-specific and effective method in Ethiopia to discuss sensitive health topics like HIV and cervical cancer screening, and could be used to start community conversations about palliative care [22, 23].

## Organization of palliative care

Our findings indicated that collaboration between different stakeholders in palliative care services was not well organized and that palliative care in Ethiopia is yet at its initiation phase. The actual practice was merely focused on medical care and financial support, rather than a holistic approach. The World Health Organization recommends palliative care programs to engage with the local health services, while health partnerships at national and regional levels are important to promote culturally safe palliative care service delivery [2]. The development of a national guideline on palliative care in Ethiopia was a good first step to align different stakeholders, however, for translation to practice much more is needed. This includes incorporating palliative care in the medical and nursing curriculum, and appoint palliative care focal persons in each regional health bureau and health facilities up to the level of health centers.

In home-based programs, caregivers are an essential component. In the present study they reported confidence in the care they provided while at the same time they asked for more training. Despite training in palliative care skills and ongoing support in the studied programs, in practice their skills did not always meet their needs and demands. The effect of peer support where caregivers can share experiences with each other could be explored to overcome their perceived lack of skills. Furthermore, home-based palliative care programs should organize care with case managers who evaluate the needs of both patient and caregiver on a regular basis and will address which needs can be alleviated by palliative care.

## Pain and symptom control

Despite participation in palliative care programs, the majority of patients who participated in our study reported moderate to severe pain. Our findings are in line with a study conducted in Addis Ababa, Ethiopia, where 65% of admitted cancer patients reported inadequate pain control [12]. Most patients in our study did use analgesics, however, data on the type, dosage, duration and adherence remained unclear and were not well-documented in the patient files. This finding raised the question whether pain was not adequately recognized or managed, or if analgesics like morphine were prescribed but not available, affordable or administered correctly. In future research, we recommend to explore pain control and its limitations more in-depth.

## Psychosocial care and the role of religion

In our study many patients reported to feel supported by their relatives and their religion. Praying provided hope and helped to get relief from their symptoms and suffering. We did not

encounter women who stopped using medication in favor of spiritual treatment. However, our cross-sectional study design among patients who are enrolled in a palliative care program might not be representative for palliative patients who do not have access to these programs. Studies in Ethiopia, Uganda, Kenya and Zimbabwe on palliative care and cervical cancer screening demonstrated that African women value support from family members and spiritual influences that play a role in their daily lives, and would prefer religious or non-pharmacological healing as compared to existing health care services [1, 24–26].

## End-of-life planning

When designing the study, we debated whether to include questions about end-of-life planning, because it is a sensitive matter to discuss and culturally sensitive interviewing is needed. In practice the question was well accepted by patients, and we found that 2 out of 5 patients had made plans, half of the patients preferred to receive palliative care at home. This is in line with a study of Reid et al conducted in Ethiopia, which reported that the majority of patients (57%) wanted to die at home [1]. However, in the study of Reid et al, the preference for dying at home was mainly reported among patients who received home-based palliative care or who experienced low pain scores. Patients with poorly controlled pain preferred in-hospital death and patients with chronic non-communicable diseases had a slight preference (58%) to die at home. In our study, all patients received home-based care and pain was not well controlled. The proportion of patients with moderate to severe pain was comparable with patients receiving home-based care in the study of Reid. The difference in preference between these patient groups might be explained by patients' experiences with in-hospital care. As mentioned previously, a study in Addis Ababa demonstrated that pain was poorly controlled in patients admitted for palliative care. This could influence patients' preference where to receive end-of-life care. Both studies illustrate it is important to organize the control of pain in palliative setting, and when this care is available at home, it can prevent unnecessary hospitalizations and crowding of healthcare facilities.

Still, the majority of patients and caregivers in our study were not involved in end-of-life planning and avoid speaking about end-of-life. Studies showed discussing about end-of-life made patients more realistic about their situation and prognosis, and reduced the likelihood of receiving intensive treatment near death [27–29].

## Preferences for home-based or institutional care

Half of the patients preferred to receive palliative care at their own home, because they consider home-based care as safe and it will ensure the presence of their family members. Other patients focused on the delivery of good quality care irrespective of the care setting. These different preferences provide opportunities for both facility and home-based care programs, depending on the patients' condition. In a setting with overburdened health facilities, home-based care could be an important service to prevent unnecessary hospitalizations.

## Limitations

The programs we assessed were different in their set-up and services, therefore challenges and strengths of one program might not be experienced by patients or caregivers in the other programs. Our recommendations, however, combine our findings from the different study settings and methods used which we believe resulted in a representative picture of the programs.

We included only female patients, because they were the target group for two (MJDA and B4G) out of the three home-based palliative care programs. Although the needs and

preferences we identified were not specific for the female role in the community or for complaints specific to female cancer, our findings cannot be extrapolated to male patients.

In patient files and project reports, data on initial management and pain scores were often missing. It was not possible to retrieve this via the patient or care provider due to recall bias. A prospective study design into the effect of palliative care on pain and the quality of life could help to answer these research questions.

## Recommendations

To understand which strategies are well-accepted and effective for patients and caregivers, we propose to conduct a prospective study in both hospital and home-based care setting with health care providers trained in palliative care and including a community-based awareness strategy. We suggest to assess the level of awareness at different levels, the number of referrals, symptom management and quality of life before, during and after program, if possible in a cluster design or stepped-wedge approach.

The Lancet Commission on Palliative Care and Pain Relief estimated that the costs for an essential package in LMIC is around 3 USD per capita [3]. When the service becomes more widely available, community health workers (in Ethiopia known as health extension workers) can inform their respective communities and refer for services. It is essential to include existing community networks, including Idirs, into the awareness strategy, in order to create a platform that is supported throughout all levels. The current unmet need combined with the increase of non-communicable diseases and the growing and aging population call for action in end-of-life care. On the other hand, structural limitations to palliative care were found evident from the study where there is no responsible structure, policy directions and guidance that could have defined who is responsible for what and how coordination could be made. This calls for more organized effort by the Federal Ministry of Health to organize responsible structure with competent human resources and financial resources at different levels. With that structure, it is critical to define types of care at household, community and facility level and defined roles and responsibilities of the different stakeholders.

## Conclusion

Our study explored palliative care needs and preferences of female patients with breast cancer, cervical cancer, HIV/AIDS or other life-limiting chronic diseases, and their caregivers in three home-based palliative care programs in Ethiopia. Patients and caregivers positively experienced the care and support provided, however, it was not sufficient. The majority of patients still suffered from moderate to severe pain and there was an unmet need in psychosocial, spiritual, economic and emotional support. Emotional and spiritual support was mainly provided by religious leaders and relatives. A minority of patients planned for the end-of-life, hoping their situation would still improve.

Considering the lack of palliative care options in Ethiopia and the challenges patients and caregivers are facing, a clear organizational structure including ongoing training and supervision of health care providers and caregivers is essential. The current practice with relatives as caregivers and home-based care calls for active involvement of community-based networks and structures. Multi-sectorial collaboration is needed to improve the quality of care and access to palliative care services.

## Supporting information

**S1 Appendix. Data collection form for patient files and project reports.**
(DOCX)

**S2 Appendix. Interview guide for patients (English version).**
(DOCX)

**S3 Appendix. Interview guide for caregivers (English version).**
(DOCX)

**S4 Appendix. Interview guide for stakeholders (English version).**
(DOCX)

**S1 Dataset.**
(RAR)

## Acknowledgments

The authors want to express their gratitude to all study participants for willingly sharing their thoughts and stories. We thank Dr Jamie Mumford and Sue Mumford of Hospice Ethiopia UK for their valuable input and comments. The authors would like to pass their gratitude to the EFCI program which is managed by Cordaid Ethiopia and supported by the Pink and Red Ribbon, Bristol Myers Squibb Foundation, the Female Cancer Foundation and Addis Ababa University School of Public Health, and to Hospice Ethiopia for supporting this study.

## Author Contributions

**Conceptualization:** Mirgissa Kaba, Marlieke de Fouw, Kalkidan Solomon Deribe, Jogchum Jan Beltman.

**Formal analysis:** Mirgissa Kaba, Marlieke de Fouw, Kalkidan Solomon Deribe.

**Funding acquisition:** Marlieke de Fouw.

**Investigation:** Mirgissa Kaba, Kalkidan Solomon Deribe, Ephrem Abathun.

**Methodology:** Mirgissa Kaba, Marlieke de Fouw, Kalkidan Solomon Deribe.

**Project administration:** Mirgissa Kaba, Marlieke de Fouw.

**Resources:** Mirgissa Kaba.

**Software:** Kalkidan Solomon Deribe.

**Supervision:** Mirgissa Kaba.

**Validation:** Mirgissa Kaba.

**Writing – original draft:** Mirgissa Kaba, Marlieke de Fouw, Kalkidan Solomon Deribe.

**Writing – review & editing:** Mirgissa Kaba, Marlieke de Fouw, Kalkidan Solomon Deribe, Ephrem Abathun, Alexander Arnold Willem Peters, Jogchum Jan Beltman.

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
