## [Decision Letter · Decision Letter 0]

29 Jun 2020

PONE-D-20-13293

Palliative care needs and preferences of female patients and their caregivers in Ethiopia rapid program assessment

PLOS ONE

Dear Dr. Deribe,

Thank you for submitting your manuscript to PLOS ONE. After careful consideration, we feel that it has merit but does not fully meet PLOS ONE’s publication criteria as it currently stands. Therefore, we invite you to submit a revised version of the manuscript that addresses the points raised during the review process.

We look forward to receiving your revised manuscript.

Kind regards,

Tim Luckett

Academic Editor

PLOS ONE

Journal Requirements:

2. When reporting the results of qualitative research, we suggest consulting the COREQ guidelines: http://intqhc.oxfordjournals.org/content/19/6/349. In this case, please consider including more information on the number of interviewers, their training and characteristics; and please provide the interview guide used.

3. We noticed you have some minor occurrence of overlapping text with the following source, which needs to be addressed:

- https://treub-maatschappij.org/2019/07/03/palliative-care-needs-and-preferences-in-ethiopia/

The text that needs to be addressed involves some sentences of the Introduction.

In your revision ensure you cite all your sources (including your own works), and quote or rephrase any duplicated text outside the methods section. Further consideration is dependent on these concerns being addressed.

4. Thank you for your ethics statement:

"The study was reviewed by the Research and Ethics committee of the School of Public Health, Addis Ababa University, and registered with number prv/154/10. "

a. Please amend your current ethics statement to confirm that your named institutional review board or ethics committee specifically approved this study.

5. Please amend either the title on the online submission form (via Edit Submission) or the title in the manuscript so that they are identical.

7. Please include captions for your Supporting Information files at the end of your manuscript, and update any in-text citations to match accordingly. Please see our Supporting Information guidelines for more information: http://journals.plos.org/plosone/s/supporting-information

8. Your ethics statement must appear in the Methods section of your manuscript. If your ethics statement is written in any section besides the Methods, please move it to the Methods section and delete it from any other section. Please also ensure that your ethics statement is included in your manuscript, as the ethics section of your online submission will not be published alongside your manuscript.

Reviewers' comments:

Reviewer's Responses to Questions

**Comments to the Author**

1. Is the manuscript technically sound, and do the data support the conclusions?

Reviewer #1: Yes

Reviewer #2: Partly

2. Has the statistical analysis been performed appropriately and rigorously? 

Reviewer #1: Yes

Reviewer #2: No

3. Have the authors made all data underlying the findings in their manuscript fully available?

Reviewer #1: Yes

Reviewer #2: Yes

4. Is the manuscript presented in an intelligible fashion and written in standard English?

Reviewer #1: Yes

Reviewer #2: Yes

5. Review Comments to the Author

Reviewer #1: This is an important paper in an understudied population. The results are very interesting, especially the combination of qualitative and quantitative results. A few minor points: the first author listed initially is different from the one listed at the start of the actual manuscript, there are grammatical errors throughout. As for layout of the qualitative results, it might be helpful to the reader if the authors used textboxes to contain a series of quoted statements along the same theme. In one of the final paragraphs, authors cite Reid et al and state that their results contradict those findings in so far as preference for death at home vs in hospital. A closer look at the Reid et al results suggests that patients with high pain scores stated a preference for a hospital-based death. As the current population also had high pain scores, I believe the two align rather than contradict each other, and furthermore that controlling pain might affect patient choice in end of life preferences, which is also interesting as far as minimizing hospital-based deaths in this resource limited setting.

Reviewer #2: Many thanks for the opportunity to read this article and very important aspect of care; palliative care, and its provision in Ethiopia.

Abstract

The study title provides no suggestion of the study design. I was not sure that this was a mixed methods study until nearly completing reading of the methods section. Please can you specify the study design either in the title or clearly in the methods section of the abstract

In terms of the sampling method described, should ‘purposely selected’ be ‘purposively sampled’?

The sentence outlining participants were “interviewed on palliative care needs and preferences, palliative care service provision and users of the service” it is not clear whether all were interviewed on all topics or whether some of these were explored with some participants.

Could you specify number of interviews with the different stakeholder groups in the abstract?

In the results of the abstract, you state pain was the main complaint and then an ordering of which other symptoms were problematic. Given this was a qualitative paper it seems unusual to present these in such a way. You could potential revise the way this is presented to suggest these were symptoms that were problematic and experienced to differing levels by participants.

Intro – clearly written and sets the context of the study well

Method

Could you provide clarity on why data gathered in 2018 is only being submitted for review at this point? Do you suspect the data will still be valid and reflect the current situation?

Would it be possible to provide data on the wider context of palliative care in Ethiopia? How reflective are participating sites? What proportion are these of all sites in the country?

Please can you outline more details about the rationale for the 10 – 15% of the total case load being the deemed a representative sample?

Pleas cite which approach to Thematic Analysis you used for analysing the interview data. What was the process for developing themes through analysis, how were discrepancies discussed?

Please specify which researchers conducted the analysis (and use initials in brackets if these were study authors)

Were caregiver participants only included if the patient they are for also participated? Or were just caregivers recruited in some instances? Who was present for interviews – were these done individually or in dyads as this may have implications for the way data is analysed

Results

The text would benefit from a proof read for English with some errors noted throughout the writing of the results section (e.g. were / was0

Table 1 – I am not sure whether the patient data regarding HIV testing relates to the patient cohort, or the patients for whom caregivers provide support – or whether these are the same. It may be helpful to readers to keep data on patient and caregiver participants separate even if presented in the same table.

Discussion

For recommendations, can you provide tangible details of findings that could be used by palliative care providers or policymakers to inform improvements in palliative care provision now?

6. PLOS authors have the option to publish the peer review history of their article (what does this mean?). If published, this will include your full peer review and any attached files.

Reviewer #1: No

Reviewer #2: No

---

## [Author Response · Author response to Decision Letter 0]

24 Aug 2020

Response to reviewers PONE-D-20-13293

1. Please ensure that your manuscript meets PLOS ONE’s style requirements, including those for file naming. 

Reply to journal requirement 1: 

We apologize for the inconvenience caused by not meeting PLOS ONE’s style requirements, and have adjusted our manuscript and file naming accordingly.

2. When reporting the results of qualitative research, we suggest consulting the COREQ guidelines: http://intqhc.oxfordjournals.org/content/19/6/349. In this case, please consider including more information on the number of interviewers, their training and characteristics; and please provide the interview guide used.

Reply to journal requirement 2: 

- Number of interviewers: there were three interviewers in Addis Ababa, two interviewers in Sidama

- Training of the interviewers: 

o Addis Ababa: all interviewers were students of the Master in Public Health, experienced with data collection in social and medical research.

o Sidama: all interviewers had at least a first degree in a health-related field and were experienced with data collection in social and medical research.

All interviewers were trained during 2 days by MK on objectives, research protocol, tools and procedures

- Characteristics of the interviewers: 

o Addis Ababa: all interviewers were Ethiopian, were fluent in Amharic, two were female, one was male

o Sidama: all interviewers were Ethiopian, were fluent in both Amharic and Sidama language, two were female, one was male

- Interview guide used: added as supplementary material S2, S3, S4 (see journal requirement 7).

The methods section has been adjusted accordingly:

The interviews were conducted by six data collectors. In Addis Ababa the team consisted of three Amharic speaking students (two female, one male) of the Master program of Public Health of Addis Ababa University. In Sidama zone the interviews were conducted by data collectors (two female, one male) fluent in both Amharic and Sidama language, with at least a first degree in a health-related field. All data collectors were recruited based on their experience in social and medical research. All data collectors were trained on the objective of the study, the research protocol including the tools and procedures of data collection for two days by the local principal investigator [MK].

3. We noticed you have some minor occurrence of overlapping text with the following source, which needs to be addressed:

- https://treub-maatschappij.org/2019/07/03/palliative-care-needs-and-preferences-in-ethiopia/

The text that needs to be addressed involves some sentences of the Introduction.

In your revision ensure you cite all your sources (including your own works), and quote or rephrase any duplicated text outside the methods section. Further consideration is dependent on these concerns being addressed.

Reply to journal requirement 3: 

The link refers to the website of the Treub Foundation (in Dutch: Treub Maatschappij), which provided financial support to facilitate the operational costs of this research. One of the conditions for funding is a short report about the research for publication on their website. Therefore there is overlapping text with the manuscript presented to you, which contains the full research description and detailed findings. Since the summary at the website does not present the methodology, detailed findings and in-depth interpretation of our study, we believe it does not constitute dual submission. We have included this information in the ‘Dual publication’ section and ‘funding section’ of Editorial Manager. 

4. Thank you for your ethics statement:

"The study was reviewed by the Research and Ethics committee of the School of Public Health, Addis Ababa University, and registered with number prv/154/10. "

a. Please amend your current ethics statement to confirm that your named institutional review board or ethics committee specifically approved this study.

Reply to journal requirement 4:

a. We have amended the statement about the ethical review, the protocol was reviewed and approved by the Research and Ethics committee of the School of Public Health, Addis Ababa.

b. According to your suggestion, we have amended the text in the ‘Ethics Statement’ field of the submission form.

5. Please amend either the title on the online submission form (via Edit Submission) or the title in the manuscript so that they are identical.

Reply to journal requirement 5: 

Thank you for noticing this inconsistency in titles, we have aligned both titles in the manuscript and the online submission form.

Reply to journal requirement 6: 

The ORCID iD of dr. Mirgissa Kaba, first author, is 0000-0002-8093-5900, and is updated in the Editorial Manager.

7. Please include captions for your Supporting Information files at the end of your manuscript, and update any in-text citations to match accordingly. Please see our Supporting Information guidelines for more information: http://journals.plos.org/plosone/s/supporting-information

Reply to journal requirement 7: 

We have updated the numbers of the Appendices in our manuscript in the correct order and according to the PLOS ONE style, and included captions for the Appendices at the end of our manuscript.

8. Your ethics statement must appear in the Methods section of your manuscript. If your ethics statement is written in any section besides the Methods, please move it to the Methods section and delete it from any other section. Please also ensure that your ethics statement is included in your manuscript, as the ethics section of your online submission will not be published alongside your manuscript.

Reply to journal requirement 8: 

The ethics statement is the last chapter of the Methods section in our manuscript. We moved the additional information about ethical clearance that was located at the end of the manuscript to the Methods section, and removed it from the end of the manuscript. (see also reply to comment number 4)

Reviewer 1 comments:

1. The first author listed initially is different from the one listed at the start of the actual manuscript.

Reply to reviewer 1 comment 1:

Thank you for noticing this mistake, the author sequence was not uploaded correctly in the system, it should have followed the sequence as presented in the manuscript. We have corrected it according to the actual manuscript. 

2. There are grammatical errors throughout.

Reply to reviewer 1 comment 2:

We apologize for the grammatical errors and have corrected the text. You will find all changes highlighted with ‘track changes’ in the revised manuscript.

3. As for layout of the qualitative results, it might be helpful to the reader if the authors used textboxes to contain a series of quoted statements along the same theme.

Reply to reviewer 1 comment 3:

Thank you for your suggestion to present our qualitative results in a clearer way to our readers. We have clustered some themes to make it more comprehensive, and included textboxes categorized by theme. 

4. In one of the final paragraphs, authors cite Reid et al and state that their results contradict those findings in so far as preference for death at home vs in hospital. A closer look at the Reid et al results suggests that patients with high pain scores stated a preference for a hospital-based death. As the current population also had high pain scores, I believe the two align rather than contradict each other, and furthermore that controlling pain might affect patient choice in end of life preferences, which is also interesting as far as minimizing hospital-based deaths in this resource limited setting.

Reply to reviewer 1 comment 4:

This is an interesting remark and perspective on the findings of Reid et al. Indeed the group with the highest pain score (patients from the oncology clinic) preferred in-hospital death above dying at home. However, the majority of patients (80%) in both the NCD clinic and home-based palliative care group reported moderate to severe pain and still preferred to die at home. This preference was more pronounced in the home-based palliative care group than the NCD clinic group (79 vs 58%). In our study, the moderate to severe pain scores were similar to the home-based and NCD clinic group, but still only half of the patients in our study preferred to die at home. This could be explained by patients experiences with the care received in clinics, compared to the care received at home. We have added the following text to the discussion section:

In our study we asked where patients preferred to receive palliative care, half of the patients preferred to receive care at home. This is in line with a study of Reid et al conducted in Ethiopia, which reported that the majority of patients (57%) wanted to die at home. However, in the study of Reid et al, the preference for dying at home was mainly reported among patients who already received home-based palliative care or who experienced low pain scores. Patients with poorly controlled pain preferred in-hospital death and patients with chronic non-communicable diseases had a slight preference (58%) to die at home. In our study, all patients received home-based care and pain was not well controlled. However, the proportion of patients with moderate to severe pain was comparable with patients receiving home-based care in the study of Reid. The difference in preference between these patient groups might be explained by patients’ experiences with in-hospital care. As mentioned previously, a study in Addis Ababa demonstrated that pain was poorly controlled in patients admitted for palliative care. This could influence patients’ preference where to receive end-of-life care. Both studies illustrate it is important to organize the control of pain in palliative setting, and when this care is available at home, it can prevent unnecessary hospitalizations and crowding of healthcare facilities.

Reviewer 2 comments:

1. ABSTRACT comment 1: The study title provides no suggestion of the study design. I was not sure that this was a mixed methods study until nearly completing reading of the methods section. Please can you specify the study design either in the title or clearly in the methods section of the abstract.

Reply to reviewer 2 ABSTRACT comment 1:

When reviewing our manuscript after receiving your comments, we understand it should be more clear from the beginning which study design we have used. Therefore we included this in the title and abstract of our revised manuscript.

2. ABSTRACT comment 2: In terms of the sampling method described, should ‘purposely selected’ be ‘purposively sampled’?

Reply to reviewer 2 ABSTRACT comment 2:

Thank you for your remark, this indeed was a mistake from our side and is corrected in the revised version.

3. ABSTRACT comment 3: The sentence outlining participants were “interviewed on palliative care needs and preferences, palliative care service provision and users of the service” it is not clear whether all were interviewed on all topics or whether some of these were explored with some participants.

Reply to reviewer 2 ABSTRACT comment 3:

We understand this sentence is confusing, because stakeholders were not asked about their palliative care needs. We have adjusted it as follows: Women who were enrolled in the palliative care programs, their primary caregiver and volunteers were interviewed on palliative care needs, preferences and service provision. We explored the views of purposely selected stakeholders on the organization of palliative care and its opportunities and challenges.

4. ABSTRACT comment 4: Could you specify number of interviews with the different stakeholder groups in the abstract?

Reply to reviewer 2 ABSTRACT comment 4:

We have adjusted the abstract accordingly: A total of 77 interviews (34 patients, 12 primary caregivers, 15 voluntary caregivers, 16 stakeholders) were conducted.

5. ABSTRACT comment 5: In the results of the abstract, you state pain was the main complaint and then an ordering of which other symptoms were problematic. Given this was a qualitative paper it seems unusual to present these in such a way. You could potential revise the way this is presented to suggest these were symptoms that were problematic and experienced to differing levels by participants.

Reply to reviewer 2 ABSTRACT comment 5:

Thank you for your request to clarify this statement. We used the African Palliative Outcome Scale (POS) to assess the presence of symptoms and its severity at the time of the interview (see Methods section). This is quantitative data and therefore we presented it in order of prevalence and severity.

6. INTRO - clearly written and sets the context of the study well

Reply to reviewer 2 INTRO comment 6: Thank you for your positive feedback.

7. METHODS - Could you provide clarity on why data gathered in 2018 is only being submitted for review at this point? Do you suspect the data will still be valid and reflect the current situation?

Reply to reviewer 2 METHODS comment 7: 

We had prepared our manuscript for publication last year and submitted it for review to a palliative care focused journal. Unfortunately, the journal was unable to find reviewers and it took time before we received this update. It was only then we were able to prepare our manuscript for a new submission. We think our data is still reflecting the current situation, the landscape of palliative care services in Ethiopia has not changed much since our research was conducted.

8. METHODS - Would it be possible to provide data on the wider context of palliative care in Ethiopia? How reflective are participating sites? What proportion are these of all sites in the country?

Reply to reviewer 2 METHODS comment 8: 

The provision of palliative care in Ethiopia is scarce. To the best of our knowledge, there is a lack of programs providing home-based palliative care. As explained in the Methods section, the programs included in this study are to our knowledge the only home-based palliative care and support programs in cancer care in Ethiopia. As stated in the results section, Black Lion Hospital, St Paul’s hospital and Yekatit Hospital in Addis Ababa provide palliative care services, but none of the hospitals has inpatient hospice care or home-based palliative care programs. Therefore, the participating sites are reflective in our opinion, although we are aware that there might be palliative care activities that we have not heard about, and we have not assessed the needs and preferences of patients that are not enrolled in any palliative care program. 

9. METHODS - Please can you outline more details about the rationale for the 10 – 15% of the total case load being the deemed a representative sample?

Reply to reviewer 2 METHODS comment 9: 

Our study reviewed the perspectives of patients and caregivers in three different settings using the rapid evaluation methodology with a mixed methods approach. It was the objective of our study to gain insight in the perspectives of patients and caregivers to create better insight in the existing services and its gaps, with the objective to formulate recommendations for the future. Unlike in primarily qualitative research, saturation of themes is not the objective of a rapid program evaluation. Looking into the total case population of each program (approximately 80-120 enrolled patients per program) of which the majority fulfilled our inclusion criteria, combined with the aim of a rapid program evaluation we aimed for approximately 8-12 patients per setting. In case we would find very different perspectives during the data collection, we decided we would extend the number of inclusions, but this was not indicated. Still, as mentioned in the discussion section, with this approach we might have missed certain themes or perspectives.

10. METHODS - Please cite which approach to Thematic Analysis you used for analysing the interview data. What was the process for developing themes through analysis, how were discrepancies discussed?

Reply to reviewer 2 METHODS comment 10:

We analyzed the interview data inductively with pre-identified codes by 2 researchers [MK, KS]. In case of discrepancies, a third researcher [MF] was involved to discuss the different opinions and reach consensus.

11. METHODS - Please specify which researchers conducted the analysis (and use initials in brackets if these were study authors)

Reply to reviewer 2 METHODS comment 11: 

We have included this information with initials in brackets do the Methods section, and the author contributions section.

12. METHODS - Were caregiver participants only included if the patient they are for also participated? Or were just caregivers recruited in some instances? Who was present for interviews – were these done individually or in dyads as this may have implications for the way data is analysed

Reply to reviewer 2 METHODS comment 12: 

We invited both patients and their caregivers for an individual interview. We have not approached caregivers separately from patients. The number of caregivers was lower than for patients, because not all caregivers were present at the time of the scheduled interview or did not provide informed consent. All interviews were conducted individually, with the patient or caregiver, and the researcher. In Yirgalem a translator was present as a third person when necessary. 

13. RESULTS - The text would benefit from a proof read for English with some errors noted throughout the writing of the results section (e.g. were / was)

Reply to reviewer 2 METHODS comment 13: 

We apologize for the grammatical errors and have corrected the text. You will find all changes highlighted with ‘track changes’ in the revised manuscript.

14. RESULTS - Table 1 – I am not sure whether the patient data regarding HIV testing relates to the patient cohort, or the patients for whom caregivers provide support – or whether these are the same. It may be helpful to readers to keep data on patient and caregiver participants separate even if presented in the same table.

Reply to reviewer 2 RESULTS comment 14: 

According to your suggestion we have prepared a separate table for the characteristics of caregivers to avoid confusion. Table 1 for patients, table 2 for caregivers.

15. DISCUSSION - For recommendations, can you provide tangible details of findings that could be used by palliative care providers or policymakers to inform improvements in palliative care provision now?

Reply to reviewer 2 DISCUSSION comment 15: 

We have adjusted our recommendations in the discussion section, in order to provide more tangible solutions to improve palliative care in Ethiopia.

---

## [Decision Letter · Decision Letter 1]

28 Sep 2020

PONE-D-20-13293R1

Palliative care needs and preferences of female patients and their caregivers in Ethiopia: a rapid program evaluation in Addis Ababa and Sidama zone

PLOS ONE

Dear Dr. Deribe,

Thank you for submitting your manuscript to PLOS ONE. After careful consideration, we feel that it has merit but does not fully meet PLOS ONE’s publication criteria as it currently stands. Therefore, we invite you to submit a revised version of the manuscript that addresses the points raised during the review process.

We look forward to receiving your revised manuscript.

Kind regards,

Tim Luckett

Academic Editor

PLOS ONE

Reviewers' comments:

Reviewer's Responses to Questions

**Comments to the Author**

1. If the authors have adequately addressed your comments raised in a previous round of review and you feel that this manuscript is now acceptable for publication, you may indicate that here to bypass the “Comments to the Author” section, enter your conflict of interest statement in the “Confidential to Editor” section, and submit your "Accept" recommendation.

Reviewer #1: All comments have been addressed

Reviewer #2: (No Response)

2. Is the manuscript technically sound, and do the data support the conclusions?

Reviewer #1: Yes

Reviewer #2: Yes

3. Has the statistical analysis been performed appropriately and rigorously? 

Reviewer #1: Yes

Reviewer #2: Yes

4. Have the authors made all data underlying the findings in their manuscript fully available?

Reviewer #1: Yes

Reviewer #2: No

5. Is the manuscript presented in an intelligible fashion and written in standard English?

Reviewer #1: Yes

Reviewer #2: Yes

6. Review Comments to the Author

Reviewer #1: (No Response)

Reviewer #2: Thank you addressing previous comments in this current revision. Most of my queries have been addressed, but it would be helpful to include aspects of your response into the body of the manuscript.

For example, thank you for clarification on the way in which you sought to capture a representative sample. However, it would be helpful for readers of the manuscript if this rationale outlined in your response was outlined in the body of the manuscript so that your decision making around accessing the 10 – 15% of the total case load is clear. Similarly, your mention of continuously monitoring the data during the project (to determine whether new themes were emerging) would be useful to add further transparency to the approach adopted.

Thank you for clarifying that you adopted an inductive approach to framework analysis. Please can you add the term ‘inductive’ into the body of the manuscript. I was, however, unclear on how pre-identified codes were used within the inductive approach. This may be an issue with terminology, but please can you provide more detail about the use of pre-identified codes.

Please add details into the manuscript regarding which caregivers were involved in the interviews, and details around interviews with patient and caregiver participants being conducted alone, as you have outlined in your response.

The recommendations are very specific, aligning with a proposed research design or an awareness strategy. Is it possible to broaden the recommendations to consider, for example, how the findings align with the pillars of palliative care (i.e. policy, education, drug availability, implementation, and research) (https://pubmed.ncbi.nlm.nih.gov/23561750/) or policy relating to provision of care in Ethiopia?

7. PLOS authors have the option to publish the peer review history of their article (what does this mean?). If published, this will include your full peer review and any attached files.

Reviewer #1: No

Reviewer #2: No

---

## [Author Response · Author response to Decision Letter 1]

12 Oct 2020

Response to reviewers PONE-D-20-13293R1 

Have the authors made all data underlying the findings in their manuscript fully available? 

The Plos data policy requires authors to make all data underlying the findings described in their manuscript fully available without restriction, with rare exception (please refer to the Data Availability Statement in the manuscript PDF file). The data should be provided as part of the manuscript or its supporting information, or deposited to a public repository. For example, in addition to summary statistics, the data points behind means, medians and variance measures should be available. If there are restrictions on publicly sharing data e.g. participant privacy or use of data from a third party—those must be specified. 

Reviewer # 2 comments: No

Reply to Reviewer 2 comments

Thank you. The data is fully accessible as supporting information and also it can be available from the corresponding author upon reasonable request.

Reviewer #2: Thank you addressing previous comments in this current revision. Most of my queries have been addressed, but it would be helpful to include aspects of your response into the body of the manuscript. For example, thank you for clarification on the way in which you sought to capture a representative sample. However, it would be helpful for readers of the manuscript if this rationale outlined in your response was outlined in the body of the manuscript so that your decision making around accessing the 10 – 15% of the total case load is clear. 

Reply to Reviewer 2 comments

Thank you for the additional suggestion to provide the rationale outlined in the response in the body of the manuscript to clarify on accessing 10-15% of the total case load. We provided the rationale in the method section (Method section, page 5&6, from line 120 to 127)

Reviewer #2: Similarly, your mention of continuously monitoring the data during the project (to determine whether new themes were emerging) would be useful to add further transparency to the approach adopted.

Reply to Reviewer 2 comments

Thank you for this useful insight. Here we were referring to the research process as project, which we realized may give a wrong impression. The point here has to do with the fact that we developed themes based on the research questions before the data collection. During the data collection as well as transcription, we kept note of emerging themes and/or if pre-defined themes still are valid based on evidences generated from the data. The analysis thus follows the themes that were finalized following completion of data collection (Method section, page 7, from line 183 to 186).

Reviewer #2: Thank you for clarifying that you adopted an inductive approach to framework analysis. Please can you add the term ‘inductive’ into the body of the manuscript. I was, however, unclear on how pre-identified codes were used within the inductive approach. This may be an issue with terminology, but please can you provide more detail about the use of pre-identified codes.

Reply to Reviewer 2 comments

Thank you again for this useful point. We fully agree this is a bit confusing. Codes were developed early on before data collection based on the research question. These codes were finalized later based on data generated and transcribed. The coding of interview data was made by two researchers [MK and KS] with support from [MF] in cases of discrepancies. While the pre-coding is possible that doesn’t warrant inductive approach. So, this is removed and corrections made accordingly (Method section, page 7&8, from line 181 to 187)

Reviewer #2: Please add details into the manuscript regarding which caregivers were involved in the interviews, and details around interviews with patient and caregiver participants being conducted alone, as you have outlined in your response.

Reply to Reviewer 2 comments

Thank you for this insight which will make the reading clearer. We included this information in the revised manuscript (Method section, page 6, from line 128 to 140. Additional details about how interviews with patient and caregiver participants being conducted has also provided in data collection and ethics sections.

Reviewer #2: The recommendations are very specific, aligning with a proposed research design or an awareness strategy. Is it possible to broaden the recommendations to consider, for example, how the findings align with the pillars of palliative care (i.e. policy, education, drug availability, implementation, and research) or policy relating to provision of care in Ethiopia? 

Reply to Reviewer 2 comments

We appreciate for this. We considered broadening the recommendation has an added value to the research outcome that we broadened the recommendations based on the study finding (Recommendation section, page 22, from line 484 to 498)

---

## [Decision Letter · Decision Letter 2]

30 Oct 2020

PONE-D-20-13293R2

Palliative care needs and preferences of female patients and their caregivers in Ethiopia: a rapid program evaluation in Addis Ababa and Sidama zone

PLOS ONE

Dear Dr. Deribe,

Thank you for submitting your manuscript to PLOS ONE. After careful consideration, we feel that it has merit but does not fully meet PLOS ONE’s publication criteria as it currently stands. Therefore, we invite you to submit a revised version of the manuscript that addresses the points raised during the review process.

We look forward to receiving your revised manuscript.

Kind regards,

Tim Luckett

Academic Editor

PLOS ONE

Additional Editor Comments:

Like Reviewer #2, I would like to question the part of your Methods where contrast your 'mixed methods' approach with a 'primarily qualitative approach'. Please provide more information about the type of mixed methods you employed according to a recognised typology, such as Creswell's (2018), including details of how quantitative and qualitative data were integrated. As they are currently written, your Methods and Results look very much like a qualitative study, where the quantitative data were used only to describe the sample and nothing more? 

On a related point, while you indicate that you did not use saturation to inform sample size, you don't suggest what alternative method was used beyond specifying what appears to be an arbitrary 10-15% 124 of the total case load of each program?

Reviewers' comments:

**Comments to the Author**

1. If the authors have adequately addressed your comments raised in a previous round of review and you feel that this manuscript is now acceptable for publication, you may indicate that here to bypass the “Comments to the Author” section, enter your conflict of interest statement in the “Confidential to Editor” section, and submit your "Accept" recommendation.

Reviewer #2: (No Response)

2. Is the manuscript technically sound, and do the data support the conclusions?

Reviewer #2: Partly

3. Has the statistical analysis been performed appropriately and rigorously? 

Reviewer #2: N/A

4. Have the authors made all data underlying the findings in their manuscript fully available?

Reviewer #2: Yes

5. Is the manuscript presented in an intelligible fashion and written in standard English?

Reviewer #2: Yes

6. Review Comments to the Author

Reviewer #2: Thank you for efforts to address my previous comments. These are now mostly resolved. Prior to making a recommendation for publication there are two remaining points that I feel need addressing first.

1) I am keen to ensure transparency in the reporting of the methods around your handling of the interview data. I appreciate that this was a rapid programme evaluation, but the methods still need to be very clear. Previous revisions have helped to clarify the approach you have taken, but following your last response (removal of the 'inductive' element) I am now unsure of which approach was adopted. Please can you clarify, was this a deductive thematic analysis that you adopted? And if so, please can you provide a supporting citation of the analysis approach, such as Braun and Clarke.

2) In your most recent response, you have outlined that the objective of your approach to interviewing was unlike other qualitative research where the focus is on achieving data saturation - I would strongly disagree with this being the only means of determining the completeness and quality of data captured in a qualitative study and would suggest that this statement is removed or revised.

These are the only remaining points and thank you for addressing previous comments.

7. PLOS authors have the option to publish the peer review history of their article (what does this mean?). If published, this will include your full peer review and any attached files.

Reviewer #2: No

---

## [Author Response · Author response to Decision Letter 2]

9 Nov 2020

Editor Comments: 

Like Reviewer #2, I would like to question the part of your Methods where contrast your 'mixed methods' approach with a 'primarily qualitative approach'. Please provide more information about the type of mixed methods you employed according to a recognized typology, such as Creswell's (2018), including details of how quantitative and qualitative data were integrated. As they are currently written, your Methods and Results look very much like a qualitative study, where the quantitative data were used only to describe the sample and nothing more?

Reply to editor comments

Thank you for this useful point. We fully agree this study is largely qualitative. Nonetheless, there are specific research questions that necessitated both qualitative and quantitative approach. we employed concurrent parallel design (Creswell and Creswell 2017) Both qualitative and quantitative data were collected in parallel and are both part of the rapid program evaluation approach as described in the method section (method section, page 5 from line 98 to 100). The quantitative data were used to describe such findings as palliative outcome scale score (POS) to express the severity of symptoms among patients, to quantify preference for institutional or home based care, to describe how many of the participants had end of life plan etc (method section, page 6, from line 152 to 155). 

Editor Comments: 

On a related point, while you indicate that you did not use saturation to inform sample size, you don't suggest what alternative method was used beyond specifying what appears to be an arbitrary 10-15% 124 of the total case load of each program?

Reply to editor comments

Thank you for your request to further specify our sample size selection. We want to clarify here that saturation was not strictly defined on how to track at the beginning. However, with 8-12 participants from the respective sites, saturation was achieved after the fifth participant was interviewed. We interviewed additional participants after presumed saturation was achieved to ascertain repetition of evidence were beyond doubt. In this research evidence was triangulated by different sources and methods, that makes the data presented in this manuscript very strong (method section, page 5 & 6, from line 121 to 136).

Reviewer # 2 comments: 

I am keen to ensure transparency in the reporting of the methods around your handling of the interview data. I appreciate that this was a rapid program evaluation, but the methods still need to be very clear. Previous revisions have helped to clarify the approach you have taken, but following your last response (removal of the 'inductive' element) I am now unsure of which approach was adopted. Please can you clarify, was this a deductive thematic analysis that you adopted? And if so, please can you provide a supporting citation of the analysis approach, such as Braun and Clarke.

Reply to Reviewer 2 comments

Thank you as well for this useful point. As pointed out above, it is true that this study is largely qualitative. Nonetheless, there are specific research question that necessitated quantitative evidence. The findings however remain descriptive that we were guided by inductive approach. Based on your previous and current comments we explained this in the body of the manuscript under method section, page 7, line 182. Qualitative data coding was developed early on before data collection based on the research question. However, these codes were refined later following transcription of data. Eventual coding of the data was made by two researchers [MK and KS] with support from [MF] in cases of discrepancies. Quantitative data helped to describe palliative outcome scale score (POS) to express the severity of symptoms among patients, to quantify preference for institutional or home based care, to describe how many of the participants had end of life plan etc that could not be addressed through qualitative method (method section, page 6, from line 152 to 155, result section, page 9 from line 233 to 241, page 12 from line 297 to 319). As it is presented in the result section qualitative data helped to explain physical signs and symptoms, perceived challenges and preferences for palliative care, end-of-life planning. perceived support, preferences for provision of palliative care, existing health service activities, gaps and challenges in palliative care programs.

Reviewer #2 comments: 

In your most recent response, you have outlined that the objective of your approach to interviewing was unlike other qualitative research where the focus is on achieving data saturation I would strongly disagree with this being the only means of determining the completeness and quality of data captured in a qualitative study and would suggest that this statement is removed or revised

Reply to Reviewer 2 comments

We agree with your insightful challenge and we revised the statement (Method section, page 5, from line 121 to 127).

Reference

Creswell, J. W. and J. D. Creswell (2017). Research design: Qualitative, quantitative, and mixed methods approaches, Sage publications.

---

## [Editor Report · Decision Letter 3]

11 Nov 2020

PONE-D-20-13293R3

Palliative care needs and preferences of female patients and their caregivers in Ethiopia: a rapid program evaluation in Addis Ababa and Sidama zone

PLOS ONE

Dear Dr. Deribe,

Thank you for submitting your manuscript to PLOS ONE. After careful consideration, we feel that it has merit but does not fully meet PLOS ONE’s publication criteria as it currently stands. Therefore, we invite you to submit a revised version of the manuscript that addresses the points raised during the review process.

We look forward to receiving your revised manuscript.

Kind regards,

Tim Luckett

Academic Editor

PLOS ONE

Editor Comments:

Thank you for attempting to address comments from the editor and reviewer on the last version of the manuscript, but unfortunately three responses were inadequate as follows:

1. Justification of why the design can be called mixed methods seems rests on there being some 'triangulation' between qualitative and quantitative components, but greater description of the process for integration is required.

2. Your response to the reviewer's request for more detail on your approach to qualitative analysis appears contradictory in that you call your approach inductive but also say that codes were predefined prior to data collection?

3. You have indicated that saturation was reached after 5 interviews, which is about half that considered more typical even for code rather than meaning saturation (Monique, 2017). Also, you had previously stated that a limitation of your study was that saturation for themes wasn't reached?

---

## [Author Response · Author response to Decision Letter 3]

5 Dec 2020

Dear Editor,

We would like to thank you for providing us with useful comments and queries. We found this a useful opportunity to improve the quality of our submission to the expected standard of your journal. Below, you will find point-by-point explanations.

Thank you again, 

Kalkidan Solomon

On behalf of all authors.

1. Justification of why the design can be called mixed methods seems rests on there being some 'triangulation' between qualitative and quantitative components, but greater description of the process for integration is required.

Thank you for this useful point. Our study reviewed the perspectives of patients, caregivers and providers in different settings using the rapid evaluation methodology with mixed method approach. It was the objective of our study to gain insight in the perspectives of service users and providers to create better insight in the existing services and its gaps, with the objective to formulate recommendations for the future. Basing this assumption, this study is largely qualitative. On the other hand, there are specific research questions that necessitated both qualitative and quantitative approach thus, we employed concurrent parallel design. Both qualitative and quantitative data were collected in parallel and are both part of the rapid program evaluation approach. As it is presented in the result section qualitative data alone helped to explain most of the study objectives including; physical signs and symptoms, perceived challenges and preferences for palliative care, end-of-life planning. perceived support, preferences for provision of palliative care, existing health service activities, gaps and challenges in palliative care programs. As it was mentioned, integration of quantitative data was necessary to describe palliative outcome scale score (POS) to express the severity of symptoms among patients, to quantify preference for institutional or home based care, to describe how many of the participants had end of life plan etc. But The findings however remain descriptive that we were guided by inductive approach. 

2. Your response to the reviewer's request for more detail on your approach to qualitative analysis appears contradictory in that you call your approach inductive but also say that codes were predefined prior to data collection?

Thank you for your suggestion. We analyzed the interview data inductively with pre-identified codes by 2 researchers [MK, KS]. In case of discrepancies, a third researcher [MF] was involved to discuss the different opinions and reach consensus. Though pre-coding is possible that doesn’t warrant inductive approach. Qualitative data coding was developed early on before data collection based on the research question. However, these codes were finalized later based on data generated and transcribed. 

3. You have indicated that saturation was reached after 5 interviews, which is about half that considered more typical even for code rather than meaning saturation (Monique, 2017). Also, you had previously stated that a limitation of your study was that saturation for themes wasn't reached?

Thank you for noticing this inaccuracy. We apologize for the errors in the limitation section and we have corrected the text. The statement which stated that “saturation for themes wasn't reached” in the limitation section was to explain about the pitfall of rapid program evaluation (REM) which we used as a method in this study. We want to clarify here that saturation was not strictly defined on how to track at the beginning since we applied rapid program evaluation methodology. However, with 8-12 participants from the respective sites, saturation was achieved after the fifth participant was interviewed. We interviewed additional participants after presumed saturation was achieved to ascertain repetition of evidence were beyond doubt (This was the actual scenario in this study). You will find all changes highlighted with ‘track changes’ in the revised manuscript (Limitation section, page 21, from line 457 to 462).

---

## [Editor Report · Decision Letter 4]

9 Dec 2020

PONE-D-20-13293R4

Palliative care needs and preferences of female patients and their caregivers in Ethiopia: a rapid program evaluation in Addis Ababa and Sidama zone

PLOS ONE

Dear Dr. Deribe,

Thank you for submitting your manuscript to PLOS ONE. After careful consideration, we feel that it has merit but does not fully meet PLOS ONE’s publication criteria as it currently stands. Therefore, we invite you to submit a revised version of the manuscript that addresses the points raised during the review process.

We look forward to receiving your revised manuscript.

Kind regards,

Tim Luckett

Academic Editor

PLOS ONE

Additional Editor Comments (if provided):

Thank you for your response, which clarifies the third question asked on my last review. However, I'm afraid you still haven't responded satisfactorily to the first and second questions. No changes have been made to clarify the method used for integration or - if no such methods were used - to change the approach from mixed methods to qualitative (noting that some survey information was also collected from participants to describe the sample). Also, your confirmation that codes were predefined prior to analysis means the approach can no longer be described as 'inductive' throughout. Might an integrated approach of the kind described by Bradley et al (2007) be more fitting, for example - https://onlinelibrary.wiley.com/doi/pdf/10.1111/j.1475-6773.2006.00684.x ?

---

## [Author Response · Author response to Decision Letter 4]

24 Dec 2020

On behalf of all authors We thank you for your constructive feedback on our manuscript entitled “Palliative care needs and preferences of female patients and their caregivers in Ethiopia; a rapid program evaluation” [PONE-D-20-13293R4]. Please find the point by point responses here below.

We trust that the improvements suggested by both academic editors and reviewers made this revised manuscript suitable for publication in PLOS ONE. We are looking forward to hearing from you. 

Yours sincerely,

Kalkidan Solomon

 Editor Comments 

1. Thank you for your response, which clarifies the third question asked on my last review. However, I'm afraid you still haven't responded satisfactorily to the first and second questions. No changes have been made to clarify the method used for integration or - if no such methods were used - to change the approach from mixed methods to qualitative (noting that some survey information was also collected from participants to describe the sample). 

 Thank you for this useful point. Our study applied rapid evaluation methodology with mixed method approach. This study is largely qualitative. On the other hand, there are specific research questions that necessitated description using quantitative approach. Both qualitative and quantitative data were collected in parallel and are both part of the rapid program evaluation approach. As it is presented in the result section qualitative data alone helped to explain vast majority of the study objectives including; physical signs and symptoms, perceived challenges and preferences for palliative care, end-of-life planning. perceived support, preferences for provision of palliative care, existing health service activities, gaps and challenges in palliative care programs. But the findings of quantitative study however remain descriptive and we didn’t have integration of both qualitative and quantitative finding for a single objective since we have used quantitative data mainly to describe/quantify palliative outcome scale score among (POS) to express the severity of symptoms among patients.

2. Also, your confirmation that codes were predefined prior to analysis means the approach can no longer be described as 'inductive' throughout. Might an integrated approach of the kind described by Bradley et al (2007) be more fitting, for example https://onlinelibrary.wiley.com/doi/pdf/10.1111/j.1475-6773.2006.00684.x ?

 Thank you for your suggestion. We analyzed the interview data inductively. As we mentioned earlier we had predefined code/ codebook prior to analysis based on the objectives but we have made a lot of amendments in to the code, we added new coded basing what we got from the data so we were free to have emerging codes from the data and all the codes, categories and themes in this study were emerging from the data itself and that is why we reported as we analyzed the interview data inductively. To avoid ambiguity in this regard and since we practically follow inductive approach we have removed the statement which stated that “predefined codes were used during analysis from the manuscript and replaced with the statement as ‘we analyzed the data inductively’ (Method section, Page 7, from line 183 to 184)

---

## [Editor Report · Decision Letter 5]

4 Jan 2021

PONE-D-20-13293R5

Palliative care needs and preferences of female patients and their caregivers in Ethiopia: a rapid program evaluation in Addis Ababa and Sidama zone

PLOS ONE

Dear Dr. Deribe,

Thank you for submitting your manuscript to PLOS ONE. After careful consideration, we feel that it has merit but does not fully meet PLOS ONE’s publication criteria as it currently stands. Therefore, we invite you to submit a revised version of the manuscript that addresses the points raised during the review process.

We look forward to receiving your revised manuscript.

Kind regards,

Tim Luckett

Academic Editor

PLOS ONE

Additional Editor Comments (if provided):

Thank you or clarifying that an inductive approach was indeed taken to the qualitative analysis reported in this manuscript, and for removing content that might be confusing to readers in this regard.

However, I am now even less satisfied that the overall approach taken can be described as mixed methods given that the authors have confirmed that no integration occurred. According to Creswell and all other leading methodologists I am aware of, integration is a requirement of mixed methods. Please either provide evidence from the literature that the approach taken meets the requirements of mixed methods without integration or else change the nomenclature throughout to describe the study as using a qualitative approach, while also collecting some quantitative data.

---

## [Author Response · Author response to Decision Letter 5]

14 Jan 2021

Response to Editor PONE-D-20-13293R5

On behalf of all authors We thank you for your constructive feedback and suggestions on our manuscript entitled “Palliative care needs and preferences of female patients and their caregivers in Ethiopia; a rapid program evaluation” [PONE-D-20-13293R5]. Please find the point by point responses here below.

We trust that the improvements suggested by both academic editors and reviewers made this revised manuscript suitable for publication in PLOS ONE. We are looking forward to hearing from you. 

Yours sincerely,

Kalkidan Solomon

 Editor Comment 

1. However, I am now even less satisfied that the overall approach taken can be described as mixed methods given that the authors have confirmed that no integration occurred. According to Creswell and all other leading methodologists I am aware of, integration is a requirement of mixed methods. Please either provide evidence from the literature that the approach taken meets the requirements of mixed methods without integration or else change the nomenclature throughout to describe the study as using a qualitative approach, while also collecting some quantitative data.

 Thank you for your suggestion. As we explained before this study is largely qualitative. By now we understand and accept your suggestion and changed the terminology throughout the manuscript to describe the study as using a qualitative approach, while also (as it is presented in the result section) the descriptive findings of quantitative study were used to describe/quantify palliative outcome scale score (POS) to express the severity of symptoms among patients. Thus we removed the content from the manuscript that might be confusing to readers in this regard. (Abstract section, Page 2 and Method Section Page 5 & 6).

---

## [Editor Report · Decision Letter 6]

15 Jan 2021

PONE-D-20-13293R6

Palliative care needs and preferences of female patients and their caregivers in Ethiopia: a rapid program evaluation in Addis Ababa and Sidama zone

PLOS ONE

Dear Dr. Deribe,

Thank you for submitting your manuscript to PLOS ONE. After careful consideration, we feel that it has merit but does not fully meet PLOS ONE’s publication criteria as it currently stands. Therefore, we invite you to submit a revised version of the manuscript that addresses the points raised during the review process.

We look forward to receiving your revised manuscript.

Kind regards,

Tim Luckett

Academic Editor

PLOS ONE

Additional Editor Comments (if provided):

Thank you for confirming that this was a qualitative study.

The manuscript will be ready for publication after some minor further edits.

Abstract

The abstract in the online system still says ‘mixed methods’ even though this has been changes to qualitative in the manuscript; please harmonise.

The methods should state that the POS was administered.

Please reword the sentence on analysis to state: ‘Descriptive analyses were used for POS data, and an inductive thematic analysis for the interview data’.

The sentence in the conclusion that begins ‘pain was poorly controlled’ really just repeats what was in the results and should be removed.

Manuscript

Please ensure ‘a’ is inserted before ‘qualitative study/approach’ throughout.

Please remove the following sentence which unnecessarily repeats: ‘Without integration with qualitative data, the quantitative data alone were used to describe palliative outcome scale score (POS) to express the severity of symptoms among patients’.

Please ensure the abbreviation POS is used throughout without the full name after this has been introduced the first time.

Cut-offs for the POS should be moved from the Results to the Methods and justified with a reference. Rather than use ‘low’ and ‘high’ which are ambiguous, I suggest just sticking with ‘none-mild’ and ‘moderate-severe’.

Please list the themes in sentence format rather than list form, i.e. ‘We identified the following themes in the semi-structured interview transcripts: awareness of palliative care; organization of palliative care and referral pathways …’

For Table 1, give the overall number of participants at the end of the title in brackets (i.e ‘(N=XX’).

Don’t repeat the contents of Table 1 in the text but only include any information that is additional.

Remove ‘age in years, median’ from Table 1 as it’s in the text and disrupts the column headings.

Move the contents of Table 2 to Table 1, as these are continuing socio-demographic characteristics.

Include an explanation in the text of why so much data were missing.

---

## [Author Response · Author response to Decision Letter 6]

28 Feb 2021

Response to Editor PONE-D-20-13293R6

On behalf of all authors We thank you for your constructive feedback and suggestions on our manuscript entitled “Palliative care needs and preferences of female patients and their caregivers in Ethiopia; a rapid program evaluation” [PONE-D-20-13293R6]. Please find the point by point responses here below.

We trust that the improvements suggested by both academic editors and reviewers made this revised manuscript suitable for publication in PLOS ONE. We are looking forward to hearing from you. 

Yours sincerely,

Kalkidan Solomon

 Editor Comment 

Abstract

1. The abstract in the online system still says ‘mixed methods’ even though this has been changes to qualitative in the manuscript; please harmonise.

Thank you. We harmonized the online content with the manuscript content

2. The methods should state that the POS was administered. 

Thank you for your suggestion. We included that POS was used to describe physical symptoms in the abstract (Abstract section, from line 38 to 40).

3. Please reword the sentence on analysis to state: ‘Descriptive analyses were used for POS data, and an inductive thematic analysis for the interview data’.

Thank you again for your suggestion. We reword and add statement which describe about POS data and inductive thematic analysis for the qualitative data (Abstract section, from line 38 to 40).

4. The sentence in the conclusion that begins ‘pain was poorly controlled’ really just repeats what was in the results and should be removed 

We accept the comment and revised accordingly. Thank you (Abstract section, from line 50 to 52)

Manuscript

1. Please ensure ‘a’ is inserted before ‘qualitative study/approach’ throughout.

Thank you for your suggestion. We just added ‘a’ before ‘qualitative study/approach’ throughout. 

2.Please remove the following sentence which unnecessarily repeats: ‘Without integration with qualitative data, the quantitative data alone were used to describe palliative outcome scale score (POS) to express the severity of symptoms among patients’.

Thank you for your feedback. We removed the statement (Method section, from line 150 to 152)

3. Please ensure the abbreviation POS is used throughout without the full name after this has been introduced the first time.

Thank you. We revised and used the abbreviation ‘POS’ throughout without the full name after POS has been introduced the first time.

4. Cut-offs for the POS should be moved from the Results to the Methods and justified with a reference. Rather than use ‘low’ and ‘high’ which are ambiguous, I suggest just sticking with ‘none-mild’ and ‘moderate-severe’.

Thank you for the valuable comments. We have moved Cut-offs for the POS to method section and added reference. In addition, ‘low and high’ words were removed to avoid confusion and as suggest we used ‘none-mild’ and ‘moderate-severe’ (Method section, from line 175 to 180 and Result section, from line 234 to 236 and Table 3, from line 626 to 629). 

5. Please list the themes in sentence format rather than list form, i.e. ‘We identified the following themes in the semi-structured interview transcripts: awareness of palliative care; organization of palliative care and referral pathways …’

We accept the suggestion and revised accordingly. Thank you (Result section, from line 244 to 256)

6. For Table 1, give the overall number of participants at the end of the title in brackets (i.e ‘(N=XX’).

Thank you for your suggestion. We add the total number of participants for both table one two (Table section, line 630 and line 637)

7. Don’t repeat the contents of Table 1 in the text but only include any information that is additional.

We accept the feedback and revised the text accordingly. Thank you (Result section, from line 220 to 228).

8. Remove ‘age in years, median’ from Table 1 as it’s in the text and disrupts the column headings.

Thank you. As suggested, we removed the row from Table 1 (Table 1, line 630).

9. Move the contents of Table 2 to Table 1, as these are continuing socio-demographic characteristics. 

Thank you for your suggestion. We presented the socio-demographic characteristics of both patients and caregivers in one table when we initially submitted this manuscript but both reviewers (#1 and 2) commented on it to split the table in to two and present the socio-demographic characteristics of the patients and caregivers in separate table then based on reviewers comment we split and presenting using two table.

10. Include an explanation in the text of why so much data were missing

Thank you for your suggestion. In this study a total of 27 care givers were included (12 with primary caregivers, 15 with voluntary caregivers) and as total number was indicated in the table we took the socio-demographic profile from all caregivers who were included in this study but we only select variables which had links with the study objectives. Nonetheless. nothing was missed while we collected the data.

---

## [Editor Report · Decision Letter 7]

5 Mar 2021

Palliative care needs and preferences of female patients and their caregivers in Ethiopia: a rapid program evaluation in Addis Ababa and Sidama zone

PONE-D-20-13293R7

Dear Dr. Deribe,

We’re pleased to inform you that your manuscript has been judged scientifically suitable for publication and will be formally accepted for publication once it meets all outstanding technical requirements.

Kind regards,

Tim Luckett

Academic Editor

PLOS ONE

---

## [Editor Report · Acceptance letter]

5 Apr 2021

PONE-D-20-13293R7 

Palliative care needs and preferences of female patients and their caregivers in Ethiopia: a rapid program evaluation in Addis Ababa and Sidama zone 

Dear Dr. Deribe:

I'm pleased to inform you that your manuscript has been deemed suitable for publication in PLOS ONE. Congratulations! Your manuscript is now with our production department. 

Kind regards, 

on behalf of

Dr. Tim Luckett 

Academic Editor

PLOS ONE